# Cell Heterogeneity and Phenotypic Plasticity in Metastasis Formation: The Case of Colon Cancer

**DOI:** 10.3390/cancers11091368

**Published:** 2019-09-14

**Authors:** Miriam Teeuwssen, Riccardo Fodde

**Affiliations:** Department of Pathology, Erasmus MC Cancer Institute, Erasmus University Medical Center, 3015 GD Rotterdam, The Netherlands; m.teeuwssen@erasmusmc.nl

**Keywords:** colon cancer, Wnt signaling, tumor heterogeneity, phenotypic plasticity, EMT, hybrid E/M, collective and single-cell migration, beta-catenin paradox

## Abstract

The adenoma-to-carcinoma progression in colon cancer is driven by a sequential accumulation of genetic alterations at specific tumor suppressors and oncogenes. In contrast, the multistage route from the primary site to metastasis formation is underlined by phenotypic plasticity, i.e., the capacity of disseminated tumor cells to undergo transiently and reversible transformations in order to adapt to the ever-changing environmental contexts. Notwithstanding the considerable body of evidence in support of the role played by epithelial-to-mesenchymal transition (EMT)/mesenchymal-to-epithelial transition (MET) in metastasis, its rate-limiting function, the detailed underlying cellular and molecular mechanisms, and the extension of the necessary morphologic and epigenetic changes are still a matter of debate. Rather than leading to a complete epithelial or mesenchymal state, the EMT/MET-program generates migrating cancer cells displaying intermediate phenotypes featuring both epithelial and mesenchymal characteristics. In this review, we will address the role of colon cancer heterogeneity and phenotypic plasticity in metastasis formation and the contribution of EMT to these processes. The alleged role of hybrid epithelial/mesenchymal (E/M) in collective and/or single-cell migration during local dissemination at the primary site and more systemic spreading will also be highlighted.

## 1. Introduction—Tumor Heterogeneity in Colon Cancer

Colon cancer is the third most commonly diagnosed malignancy and the second leading cause of cancer-related death worldwide. It is predicted that its mortality burden will increase by 75% by 2040 [1]. Apart from its clinical impact, colon cancer also represents a unique study model to elucidate the cellular and molecular mechanisms underlying tumor onset, progression towards malignancy, and metastasis formation at distant organ sites [2].

It is generally accepted that primary colon carcinomas are heterotypic, i.e., they feature a heterogeneous composition of epithelial cancer cells intermingled with lymphocytes, stromal fibroblasts, endothelial, and other cell types from the micro- and macro-environment [3]. This heterogeneity is matched by the diversity of parenchymal cancer cells encompassing a broad spectrum of morphologies, gene expression profiles, and functional characteristics [4,5,6]. Likewise, heterogeneity within the stromal compartment, i.e., the tumor microenvironment, has also been demonstrated [5,7].

Intrinsic, i.e., (epi)genetic, as well as extrinsic factors, such as spatial location within the tumor (e.g., at the invasive front vs. tumor center), inflammation, and treatment history underlie the observed intra-tumor heterogeneity. Consequently, different cellular subpopulations within the primary tumor mass and its metastatic lesions are observed [8,9]. Next to ‘spatial’ heterogeneity, ‘temporal’ heterogeneity has also been demonstrated relative to changes in the (epi)genetic landscape of colon cancer within individual tumors over time [10]. Of note, tumor heterogeneity is thought to underlie the disappointing results of many currently employed anti-cancer therapies as it not only supports tumor progression and metastatic dissemination but it also lies at the basis of the development of therapy resistance and of overall poor clinical prognosis [11].

Metastasis formation is a process encompassing multiple steps: (1) Local tumor invasion across the basement membrane into the surrounding stroma, (2) intravasation into the vasculature, (3) survival in the circulatory system, (4) extravasation into the parenchyma of the distant organ, (5) colonization into a distal organ, and (6) re-initiation of proliferation to form macroscopic metastases [12]. In order to successfully complete this challenging series of events, the most important feature of the metastasizing cancer cell is the capacity to adapt to the ever-changing environmental contexts by undergoing reversible changes in its cellular identity. This ‘*Dr. Jekyll and Mr. Hide*’ feature of migrating cancer cells is often referred to as phenotypic plasticity [13] and is controlled by epigenetic mechanisms which regulate, among other processes, epithelial-to-mesenchymal transition (EMT) and the reverse mesenchymal-to-epithelial transition (MET) [14]. 

A variety of chromatin remodeling complexes such as Polycomb and NuRD, play a central role in the transcriptional regulation of EMT-related transcription factors (EMT-TFs) and micro RNAs (miRs) by determining the accessibility of regulatory DNA elements and positioning of nucleosomes [15,16]. In addition, post-translational histone modifications which modulate chromatin folding and influence recruitment of regulatory proteins and control gene expression [17]. Accordingly, contextual EMT-promoting signals epigenetically modify the repression of epithelial genes and consequently drive the transition of cells into more mesenchymal-like states. These are epigenetically sustained unless the presence of EMT-promoting signals is discontinued leading to the reversion to more epithelial phenotypes [15].

Notwithstanding the considerable body of evidence in support of the role played by EMT/MET in metastasis, its rate-limiting function, and the detailed underlying cellular and molecular mechanisms, and the extension of the necessary morphologic and epigenetic changes are still a matter of debate [14,18,19]. Rather than leading to a complete epithelial or mesenchymal state, the EMT/MET programs generate migrating cancer cells displaying intermediate phenotypes featuring both epithelial and mesenchymal characteristics. These hybrid E/M cancer cells have been the focus of much attention in the most recent scientific literature as they are likely to be metastable and as such very efficient in causing metastasis [20].

Here, we will address the role of tumor cell heterogeneity and phenotypic plasticity in colon cancer metastasis formation and the contribution of EMT to these processes. The alleged role of hybrid E/M in collective and/or single-cell migration during local dissemination at the primary site and more systemic spreading will be highlighted.

## 2. The Adenoma-Carcinoma Sequence in Colon Cancer: The β-Catenin Paradox 

Colon cancer arises and progresses through a well-defined series of histologic stages along which normal colonic epithelial cells transform in stepwise fashion into precursor lesions which eventually evolve to increasingly more invasive and malignant stages. This sequence, often referred to as ‘the adenoma-carcinoma sequence’, features a gradual accumulation of genetic alterations in specific tumor suppressors and oncogenes generally regarded as the main underlying and driving forces in the progression of colonic adenomas towards malignancy [21].

The initiating and rate-limiting event in the vast majority of sporadic colon cancer cases is represented by the constitutive activation of canonical Wnt signaling through loss of function mutations at the *APC* (adenomatous polyposis coli) tumor suppressor gene. Alternatively, gain of function or ‘activating’ mutations in Wnt agonists such as the β-catenin (*CTNNB1*) oncogene have functionally equivalent consequences, i.e., the ligand-independent and constitutive signaling activation of the pathway [2]. The reason for the pivotal role of the Wnt/β-catenin signal transduction pathway in colon cancer onset mainly resides in its functional role in the intestinal crypt of Lieberkühn where it regulates the homeostatic equilibrium between stemness, proliferation, and differentiation [22]. In the bottom third of the crypt, where stem cells reside, Wnt signaling is particularly active due to signals from the surrounding stromal environment. Moving along the crypt-villus axis however, Wnt is progressively less active in a decreasing gradient inversely proportional to the grade of differentiation of the epithelial lining [23]. Here, in the absence of canonical Wnt ligands such as Wnt3a, intracellular β-catenin levels are controlled by the formation of a multiprotein “destruction complex” encompassing protein phosphatase 2A (PP2a), glycogen synthase kinase 3 (GSK3β) and casein kinase 1α (CK1α), and the scaffold proteins adenomatous polyposis coli (APC), and Axin1/2. This complex binds and phosphorylates β-catenin at specific serine and threonine residues, thereby targeting it for ubiquitination and proteolytic degradation by the proteasome [23] (Figure 1a). In the presence of Wnt ligands instead, i.e., in the stem cell compartment, co-activation of the Frizzled and LRP5/6 (low-density lipoprotein receptor-related proteins) receptors prevents the formation of the destruction complex thus resulting in the stabilization and consequent translocation of β-catenin from the cytoplasm to the nucleus. Here, β-catenin interacts with members of the TCF/LEF family of transcription factors and modulates the expression of a broad spectrum of Wnt downstream target genes with cellular functions ranging from stemness to proliferation [23] (Figure 1a). Consequently, loss- and gain-of-function genetic alterations in *APC* and β-catenin respectively, result in the constitutive signaling of β-catenin to the nucleus [2].

This genetic model predicts that the vast majority of colon cancers, initiated by the constitutive activation of Wnt signaling, should feature nuclear β-catenin localization throughout the entire tumor mass. However, extensive immunohistochemical analysis of sporadic colon cancers has contradicted this prediction. In fact, only a minority of colon cancer cells, non-randomly distributed along the invasive front of the primary mass and of quasi-mesenchymal morphology, show nuclear β-catenin accumulation. In contrast, the majority of more differentiated (epithelial-like) tumor cells localized inside the tumor mass are characterized by an apparently normal (membrane-bound) subcellular distribution of β-catenin together with increased cytoplasmic staining [25] (Figure 1b). This “β-catenin paradox” is generally explained by the fact that the *APC* and β-catenin mutations are necessary for the constitutive activation of the pathway though insufficient for nuclear β-catenin accumulation and full-blown Wnt signaling [24] (Figure 1b). The latter is only achieved in colon cancer cells located at the invasive front where they are exposed to stromal cues capable of further promoting the nuclear translocation of β-catenin from the cytoplasm [26]. 

Of note, the same heterogeneous β-catenin distribution, with nuclear staining in less differentiated cells located in closer proximity to the microenvironment and membranous staining in more differentiated cells in the center of the lesion, has also been observed in colon cancer metastases [27]. The reacquisition of epithelial features at the metastatic sites is required for cancer cell proliferation, as mesenchymal-like cells are generally hindered in their proliferative activity and are therefore not able to underlie the expansion of the metastasis.

Hence, different levels of Wnt signaling activity between the tumor center and the invasive front are likely to account for the ‘spatial’ intra-tumor heterogeneity and to underlie distinct Wnt downstream cellular effectors such as proliferation and EMT leading to tumor growth and invasion, respectively [28]. These observations have led to the hypothesis according to which, apart from its role in colon cancer initiation, Wnt signaling and the consequent downstream EMT activation, also underlies the onset of migrating cancer stem cells (mCSC) at the invasive front of the primary lesion which locally invade the tumor microenvironment and eventually form distant metastases [29].

This paracrine—and presumably epigenetic—control of local invasion and metastasis also offers an explanation to the so-called “progression puzzle” [30], i.e., the lack of main genetic and expression differences between matched primary tumors and metastases as reported in colon cancer and other tumor types [31,32,33]. This suggests that although the adenoma-carcinoma progression at the primary site is clearly driven by the sequential accumulation of genetic mutations at key genes, the multistage route from dissemination into the tumor microenvironment to metastasis formation is underlined by phenotypic plasticity, i.e., the capacity of circulating tumor cells (CTCs) to undergo transient phenotypic changes to adapt to the ever-changing cellular contexts en route to distant organ sites. As previously and eloquently proposed by Thomas Brabletz and collaborators, EMT and its reverse program MET play pivotal roles in regulating phenotypic plasticity of CTCs [29]. 

In the next section, we will discuss the current understanding of the role of EMT in local invasion and metastasis. 

## 3. Epithelial to Mesenchymal Transition in Local Invasion and Metastasis 

As pointed out in the previous section, tumor cells within primary and metastatic tumor masses, as well as CTCs, display substantial phenotypic heterogeneity representing various intermediate stages of the EMT program [34,35]. EMT is a developmental program exploited by carcinoma cells to switch from their epithelial state, featuring cell–cell contacts and apical–basal polarity, to more motile and invasive quasi-mesenchymal phenotypes with spindle-like morphology and front-back-end polarity. During cancer invasion, EMT provides cells with the ability to produce, interact with, and digest the surrounding extracellular matrix (ECM), detach from the primary tumor, and invade into the surrounding tissue [14]. In addition to promoting cellular migration and invasion, the transient phenotypic changes associated with formation of the mesenchymal state during EMT have been associated with the acquisition of stem-like properties, resistance to therapy, and immune suppression [36,37,38,39]. The epigenetic, and as such reversible nature of EMT is crucial as the reverse mesenchymal-to-epithelial (MET) process allows migrating cancer (stem-like) cells to regain proliferative and epithelial characteristics to colonize distant organ sites [14]. The initiation and execution of EMT are orchestrated by a set of transcription factors (i.e., *ZEB1/2*, *SNAIL1/SLUG*, and *TWIST1/2*) and miRNAs (e.g., the miR200 family) [40]. Hallmarks of EMT include the silenced expression of integral members of epithelial cell adhesion structures such as adherens- and tight-junctions, and desmosomes, and/or proteins involved in cytoskeleton (re)organization and in cell-matrix adhesion. Next, EMT-TFs can also activate the expression of mesenchymal cell markers resulting in changes in cell morphology, enhanced migratory properties, and ECM remodeling. EMT is induced by cytokines and growth factors secreted from the tumor microenvironment in response to metabolic changes, hypoxia, innate and adaptive immune responses, and treatment by cytotoxic drugs [40]. In addition, the mechanical composition and properties of the ECM also play an important role in EMT regulation. Both shear stress of cancer cells and increasing matrix stiffness in the microenvironment activate EMT, tumor invasion and metastasis [41,42,43]. In turn, as noted before, EMT also stimulates the composition and mechanics of the ECM, thereby forming a tightly controlled feedback loop that is often dysregulated in cancer.

As mentioned above, colon carcinomas display nuclear β-catenin accumulation at the invasive front simultaneously with the acquisition of mesenchymal-like morphologic features [24]. In this respect, it has been shown that EMT can be activated downstream of canonical Wnt/β-catenin signaling as GSK3β kinase activity inhibition stabilizes SLUG, thereby initiating EMT [44]. Alternatively, active Wnt signaling also inhibits SNAIL1 phosphorylation, leading to increased protein levels of this transcriptional repressor of E-cadherin, EMT initiation, and local invasion [45]. In colon cancer, overexpression of the Wnt ligand Wnt3a is associated with EMT and cancer progression. Accordingly, Wnt3a overexpression in both in vitro and in vivo models was shown to induce *SNAIL* expression thus promoting EMT, an effect that is abrogated by the Wnt antagonist Dickkopf1 (Dkk1) [28].

More recently, the intestinal microbiome has also been shown to contribute to EMT. A variety of enterotoxins secreted by microbes, including *Bacteriodes fragilis*, *Fusobacterium nucleatum,* and *Enterococcus faecalis* have been demonstrated to alter normal cell–cell adhesion by interfering with E-cadherin function [46,47,48]. *F. nucleatum* adheres through FadA (*Fusobacterium adhesion A*), an adhesion protein, to E-cadherin in colon cancer cells. The FadA/E-cadherin interaction leads to activation of β-catenin signaling and of oncogenic and inflammatory responses [48]. Interestingly, *Fusobacterium* and its associated microbiome (including *Bacteroides*, *Selenomonas*, and *Prevotella*) are sustained in distal metastases and mouse xenografts of primary colorectal tumors. Treating tumor-bearing mice with the antibiotic metronidazole reduced the amount *Fusobacterium* and abrogated cancer cell proliferation and growth [49].

### 3.1. Role of EMT in Metastasis under Debate

EMT is often defined by the respective down- and up-regulation of epithelial (e.g., E-cadherin, catenins, and cytokeratins) and mesenchymal markers (e.g., vimentin, fibronectin, and N-cadherin) by the above-mentioned *ZEB1/2*, *SNAIL1/SLUG*, and *TWIST1/2* transcription factors (EMT-TFs). However, no single transcription factor (TF) or downstream target can universally define EMT throughout different cancer types and cellular contexts. Distinct EMT-TFs are likely to act in a tumor- and dosage-specific manner and as such differentially repress or enhance the transcription of specific downstream target genes. From this perspective, the recent debate on whether EMT is an essential requirement for metastasis to occur [14,50] reflects the complexity of the network of transcription factors and their downstream targets in the activation of the EMT program. Two provocative studies, in particular, have raised questions on the relative importance of the role played by EMT along the multistep sequence of events leading to metastasis. Fischer et al. (2015) employed in vivo mesenchymal GFP reporters to study EMT onset in the MMTV-PyMT mammary cancer model. Notwithstanding the observed mesenchymal expression within the primary lesions, albeit in low proportion, and its enrichment in CTCs, GFP-positive tumor cells did not contribute to distant metastases [18]. Moreover, *Zeb1/2* inhibition by miR-200 overexpression did not reduce lung metastasis incidence. In a second study by Zheng et al. (2015) it was shown, by taking advantage of a pancreatic ductal carcinoma mouse model, that genetic ablation of *Snail1* or *Twist1* did not affect dissemination and lung metastasis development [19]. The latter is in contrast with a later study by Krebs et al. (2017), showing that *Zeb1* downregulation in the same pancreatic cancer models negatively affects the formation of precursor lesions, tumor grading, invasion, and metastasis [51]. Additionally, other studies using different cancer mouse models point to a key role of *Snail1*- and *Twist1-*driven EMT in metastatic colonization [52,53,54].

Although compelling, the Fischer et al. (2015) and Zheng et al. (2015) studies are mainly based on the analysis of individual transcription factors and downstream targets in specific tumor models [39,40] and cannot as such be used to discard EMT’s role in metastasis against an overwhelming body of experimental evidence from the scientific literature. Several TFs are known to cooperate in eliciting EMT and in controlling the extension of the execution of the trans-differentiation program. Also, EMT-TFs are known to act in a cooperative and context-dependent fashion, and loss of individual factors in specific organ sites may well not suffice to initiate EMT and facilitate metastasis formation. The same is true for the employed mesenchymal markers the expression of which cannot be employed as universal readouts of EMT activation [14,50].

### 3.2. Hybrid E/M Phenotypes and Partial EMT: Many Shades of Gray

As mentioned above, the transient and reversible nature of EMT represents an essential feature for a metastatic lesion to develop [52,54,55]. Recent experimental evidence indicates that EMT, rather than acting as a binary switch where cells transit between fully epithelial and mesenchymal states, generates a broad spectrum of intermediate E/M stages where cells co-express both types of markers [14] (Figure 2). These partial EMT states are metastable and as such confer to the cancer cell enhanced phenotypic plasticity, an essential hallmark of the migrating/metastatic cancer (stem) cell [14]. 

Two recent studies, in particular, have highlighted the relevance of partial EMT in metastasis. In a mouse model of pancreatic ductal adenocarcinoma (PDAC), Aiello et al. (2018) sorted primary tumor cells according to their membranous expression of E-cadherin (*Cdh1*). Additional RNAseq and protein analysis of *Cdh1-*negative cancer cells revealed the presence of two distinct groups of tumors: while the first resulted from the transcriptional downregulation of E-cadherin (and of other epithelial markers), the second and major group showed E-cadherin expression both at the mRNA and protein levels. However, rather than being presented at the membrane, E-cadherin was internalized in recycling endosomes [34]. These two distinct E-cadherin negative and EMT-competent subpopulations of tumor cells were also earmarked by different invasive and metastatic behavior. Whereas cancer cells featuring a complete EMT (i.e., E-cadherin downregulated at the transcriptional level) invaded the tumor microenvironment mostly as single cells, cells with internalized E-cadherin in a partial EMT state (E/M) migrate collectively as multicellular clusters which are also found in the blood of the pancreatic cancer mouse model [34]. Of note, it has also been shown that the different degrees of epithelial-mesenchymal plasticity affect the tumor cells’ metastatic organotropism, i.e., their capacity to metastasize a spectrum of different organ sites [56].

In a second study, Pastushenko et al. (2018) employed a mouse model of squamous cell carcinoma (SCC) and, by taking advantage of the different expression levels of the CD106, CD61, and CD51 cell-surface markers, identified six distinct EpCAM-negative tumor cell subpopulations, each characterized by a different degree of EMT. The different SCC subpopulations, encompassing both fully mesenchymal (complete EMT) and hybrid E/M subtypes (partial EMT), were characterized by distinct chromatin landscapes and gene expression profiles. Similar EMT-heterogeneity was also found in mouse models for metaplastic and luminal breast cancer [35]. Although the tumor-propagating capacity of hybrid E/M EpCAM-negative SCC cells was found to be comparable with that of their fully mesenchymal equivalents, those with a partial EMT phenotype showed increased CTC multiplicities and metastasis formation at distant organs [35]. Overall, partial EMT seems to confer increased phenotypic plasticity to the cancer cells especially when it comes to regaining epithelial characteristic (by MET), an essential requirement for metastasis formation at specific organ sites [14]. Of note, HNSCC (head and neck squamous cell carcinoma) cells with partial EMT are preferentially localized at the invasive front of the primary tumors in close proximity to CAFs (cancer-associated fibroblasts) [57], reminiscent of the “β-catenin paradox” in colon cancer [24]. 

The elucidation of the molecular mechanisms underlying partial EMT is still in its early days. Nonetheless, the different intermediate E/M phenotypes are likely to be driven by specific epigenetic and transcriptional modifications. Kröger et al. (2019) isolated subpopulation tumor cells stably residing in a hybrid E/M state from both in vitro and in vivo models using a human immortalized and transformed mammary epithelial cell line. These E/M tumor cells were characterized by upregulation of the SNAIL EMT-TF and of canonical Wnt-signaling. Ectopic *ZEB1* expression resulted in a fully mesenchymal transformation of the E/M cells accompanied by a reduction of their tumorigenic potential and a switch from canonical to non-canonical Wnt signaling [58].

Apart from SNAIL, other transcription factors including NUMB, GRLH2, and OVOL have been proposed to act as ‘*phenotypic stability factors*’ which promote, control, and stabilize the hybrid E/M state, possibly by interfering with the core EMT decision-making circuit [59,60]. 

As mentioned above, the cancer cell’s ability to revert back from EMT-induced phenotypes is critical for metastasis formation in distant organs and full mesenchymal transformation may result in the irreversible loss of MET capacity [58,61,62]. For example, activation of TGF-β signaling triggers EMT in carcinoma cells in a dosage-dependent fashion. Upon short-term treatment, the induced EMT is reversible. However, prolonged exposure of cancer cells to TGF-β result in more stable and irreversible transitions even upon ligand withdrawal [62]. 

Next to the specific expression signatures of EMT-related transcription factors and their downstream signaling pathways driving hybrid E/M and fully mesenchymal states in cancer cells, the existence of other alternative EMT-programs with distinct outcomes has been proposed [34,56]. 

Overall, it is still unclear whether hybrid E/M cells represent a metastable population or are just captured in a time frame transitioning from the epithelial to mesenchymal phenotype. Also, it remains uncertain which context-dependent environmental factors and downstream signaling paths are responsible for driving heterogeneous phenotypic fates during tumor progression. Nonetheless, as mentioned earlier, ample experimental evidence clearly indicates that the hybrid E/M cells state is involved in the collective invasion, migration, and dissemination of tumor cells en route to form distant metastases. In the next sections, we will portray the role of EMT in collective cell invasion into the local tumor stroma and dissemination as CTC-clusters, when compared with single migrating cancer cells that complete the full EMT-program. 

## 4. Single versus Collective Cell Migration 

The initial detachment of the cancer cell from the primary mass and its invasion in the surrounding stromal microenvironment represent critical and rate-limiting steps in the metastatic cascade responsible for 90% of deaths in patients with malignancies [12,63]. In order to invade, cancer cells employ distinct invasion modalities: single (amoeboid or mesenchymal invasion) and collective cell migration. Of note, cancer cells can switch between these invasion modes, an important feature when it comes to the development of anti-invasive and anti-metastatic therapies [64].

### 4.1. Single Cell Migration 

Cancer cells lacking interactions with neighboring tumor cells can detach from the primary mass and migrate individually into the microenvironment. There are two different mechanisms of single-cell invasion, namely amoeboid and mesenchymal migration [64]. The involvement of one of these two modes is dependent on the rigidity of the cell-matrix adhesions, the tumor cell’s capacity to remodel the extracellular matrix, and the contractility of the cytoskeleton [65]. In amoeboid invasion, an EMT-independent mechanism, cancer cells have a characteristic rounded cell shape. Here, migration relies on the contractility of cortical actomyosin, promoted by the Rho/ROCK signaling pathway [66]. The proteolysis-independent actomyosin contractility results in membrane blebbing, i.e., the formation of membrane protrusions that enable cancer cells to squeeze through gaps within the ECM [66,67]. In contrast, during mesenchymal single-cell invasion, cells adopt an elongated spindle-like phenotype with front-back polarity as a result of EMT [68,69]. Additionally, cells that engage the mesenchymal mode are dependent on the activity of enzymes such as matrix metalloproteinases (MMPs) and serine protease seprase that degrade the ECM and, as tumor cells invade, progressively create channels which can be used for the cells lagging behind the leading ones [70]. Interestingly, inhibition of ECM remodeling leads to amoeboid migration with cancer cells squeezing through pre-existing pores by actomyosin contractility [67]. Of note, MMPs are generally regarded as integral members of the EMT program. In hepatocellular carcinoma (HCC), upregulation of the EMT-TF Snail not only repressed E-cadherin transcription but also increased expression of MMP-1, MMP-2, MMP-7, and MT1-MMP leading to accelerated invasion [71,72]. Alternatively, several ECM components and even MMPs can, in some cases, act as EMT initiators [73,74]. Induction of MMP-3, also known as stromelysin-1 (SL-1), in the mammary epithelium resulted in cleavage of E-cadherin leading to removal of E-cadherin and catenins from adherens junctions, downregulation of cytokeratins, upregulation of vimentin and of endogenous MMPs [73].

Although single-cell invasion is linked to tumor cells undergoing the full EMT-program leading to suppression of E-cadherin and induction of vimentin [68,69], there is evidence that partial EMT, i.e., the retention of epithelial features, can also feature single-cell migration [75,76]. Additionally, cancer cells can switch between amoeboid and mesenchymal states spontaneously or through changes in ECM composition [67].

### 4.2. Collective Cell Migration and the Role of EMT 

In collective cell migration, cancer cells retain intact cell–cell adhesions while invading the tumor microenvironment, the vasculature, and distant organ sites [77]. A variety of migration modalities feature collective cell migration, ranging from narrow linear connected cell strands to broad sheets or compact cluster/budding of cells [77]. Unlike single-cell migration resulting from fully mesenchymal cells, the role of EMT in collective migration is subtler. Recently, using a *Drosophila melanogaster* model of colon cancer, it was shown that the Snail homolog *Sna* can activate partial EMT in tumor cells leading to their collective invasion through the basement membrane and muscle fibers [78]. Additional evidence pointing at the correlation between hybrid E/M and collective cell migration lies in the onset of ‘leader’ cells at the invasive margin that are selected to guide other ‘following’ cancer cells [79]. These leader cells show a bi-phenotypic state with mesenchymal features as altered polarity and development of protrusions at their front. Yet, they also maintain attachments to their follower cells at their rear end. The follower cells, on the other hand, retain apical–basal polarity and migrate taking advantage of the pulling force generated by leader cells [80]. Knockdown of the epithelial marker cytokeratin 14 in leader cells is sufficient to block collective migration suggesting that the hybrid E/M state is mandatory for establishment of the leader cells [79]. The onset, activity, and maintenance of leader cells are coordinated by environmental stimuli, i.e., the local increase of compression [81], soluble factors, and chemokines [82], but is also controlled within the collective tumor group by autocrine or juxtacrine fashion. Of note, also in this case several MMPs are expressed at the leading edge to facilitate ECM degradation and to create a migration path for the cell clusters [83].

Notably, non-cancer cells can also contribute to collective cell migration. The movement of cancer cells can be conducted by migratory stromal cells such as fibroblasts [84,85]) or macrophages [86,87]. Labernadie et al. (2017) demonstrated that cancer-associated fibroblasts (CAFs) exert a physical force on cancer cells that leads to their collective migration. This intercellular force transduction is achieved by the formation of heterophilic adhesion complexes between N-cadherin on the CAF membrane and E-cadherin on the cancer cell membrane [85]. Moreover, CAFs are also a source of ECM-degrading proteases such as MMPs thereby creating micro tracks used by cancer cells to migrate through [84]. In addition to degrading the ECM, CAFs also secrete growth factors and chemokines that generate chemotactic gradients to direct cell migration [88]. Last, cancer cells can ingest exosomes secreted by CAFs thereby activating intracellular pathways known to trigger EMT [89]. In colon cancer, CAFs release exosomes containing miR-92a-3p and promote invasion and chemotherapy resistance. miR-92a-3p directly binds to FBXW7 and MOAP1 thereby activating Wnt-induced EMT and mitochondrial apoptosis [89].

Overall, single and collective cell migration share some of the underlying mechanisms (e.g., cell–cell and cell-matrix communication, and the establishment of a migratory polarity). Moreover, during invasion tumor cells can switch between different modes of migration depending on intrinsic (cell adhesion) and extrinsic cues (ECM composition and density). In general terms, a complete EMT is associated with single-cell migration, whereas collective cell migration seems to result from partial EMT. Nonetheless, the mechanisms underlying the role of EMT in determining the invasion modalities, the intercellular communication among invading cells, and the tumor microenvironmental cues leading to collective migration are yet poorly defined. This is further complicated by the fact that invasion modalities are likely to be cell type-, tissue-, and time-dependent. The plasticity of cancer cells to switch between different invasion modes is a key feature and a putative target for the development of novel therapeutic strategies [90].

## 5. Circulating Tumor Cells

Circulating tumor cells (CTCs) are defined as those cancer cells disseminated from the primary tumor mass and intravasated into blood vessels which are thought to underlie metastasis at distant organ sites [91]. CTCs have been identified at different multiplicities in many carcinomas including colon, breast, prostate, lung, bladder, and gastric cancer, while they are extremely rare in healthy individuals or in patients with non-malignant disease [91]. However, even in cancer patients, CTCs are extremely rare and, accordingly, their prospective isolation and characterization have proven to be a challenge [91]. Heterogeneity also exists among CTCs, possibly reflecting the above discussed intra-tumor heterogeneity. Likewise, the existence of both single CTCs, as well as CTC clusters comprising multiple (from few to hundreds) cells, has been well established in the scientific literature [92,93]. Of note, CTC clusters are not exclusively composed of epithelial cancer cells but are often intermingled with immune cells, cancer-associated fibroblasts, tumor stroma, and platelets [94,95,96,97,98]. In addition to this heterogeneity, CTCs and CTC clusters have been captured that express both epithelial and mesenchymal features [93,99,100,101]. 

### 5.1. Single CTCs versus CTC Clusters

Single CTCs disseminate into distant organs upon EMT [14]. However, the discovery of CTC clusters has raised questions on the relative role of EMT in local invasion and systemic dissemination from the primary tumor mass. CTC clusters are defined as a group of 2–3 or more tumor cells that travel as a group through the bloodstream [91]. In 1954, Watanabe showed that, by injecting bronchogenic carcinoma cells in the jugular vein of recipient mice, tumor clumps, in contrast to single cells, were able to form metastasis [102]. Accordingly, aggregated colon cancer cells also showed increased metastatic efficiency in the liver when compared with single cells after intra-portal injection in rat [103]. These initial observations, however, did not explain how and where CTC clusters are formed. More recently, it has been demonstrated that CTC clusters do not derive from the intravascular aggregation of single CTCs or from proliferating single CTCs, but rather from clumps of primary tumor cells that collectively detach from the primary mass and enter the vasculature as CTC clusters [104,105,106]. Moreover, it was also shown that the metastatic capacity of CTC clusters was up to fifty-fold higher when compared with single CTCs [104]. Genome-wide single-cell DNA methylation analysis demonstrated distinct methylomes between CTC clusters and single CTCs in human breast cancer patients. CTC clusters were shown to be hypo-methylated at stemness- and proliferation-associated transcription regulators including OCT4, NANOG, SOX2, and SIN3A, and hyper-methylated at Polycomb target genes [107]. Lastly, the presence of circulating tumor micro emboli in peripheral blood of patients with cancer arising from colon, breast, and lung was predictive of poor survival [104,108,109].

### 5.2. CTC Cluster Heterogeneity 

The heterogeneous composition of CTC clusters encompassing parenchymal cancer cells together with immune cells, cancer-associated fibroblasts, tumor stroma, and platelets, seems to reflect the heterogeneity of the primary tumors they originate from [94,95,96,97,98]. The presence of non-malignant cells within CTC clusters contributes to their improved survival and metastatic capacity. Normal epithelial cells undergo cell anoikis upon the detachment from the extracellular matrix (ECM), which establishes an important defense mechanism to prevent abnormal growth in inappropriate places. However, EMT can circumvent anoikis in individual cells during dissemination and metastasis [110]. The transition of single CTCs to a mesenchymal phenotype results in the expression of adherence-independent survival signals that compensate for the loss of attachment to the ECM [111]. Alternatively, CTC clusters may prevent tumor cell anoikis by retaining epithelial cell–cell interactions and thus contributing to the activation of survival stimuli [104,105]. Next, the non-malignant cell microenvironment can protect CTC cells from immune cells [112,113], shield cells from mechanical stress, and promotes adhesion to the endothelium [114,115]. It also has been shown that platelets can induce EMT in CTCs via TGF-β and NF-κB signaling while enhancing their metastatic potential [97]. Thus, secretion of growth factors and cytokines by the non-cancer cells may represent an additional survival advantage for the CTC clusters in the vasculature. Last, yet another advantage of the CTC clusters when compared to single CTCs is the capacity of remodeling the microenvironment at the metastatic site, thereby facilitating colonization [95].

### 5.3. EMT in CTC Clusters

Next to the heterogeneity of CTC clusters in terms of cell lineage composition, the degree of EMT activation among the parenchymal cancer cells within CTCs can also vary considerably. CTC clusters display epithelial cell–cell interactions as shown by the retention of expression of several epithelial-specific genes such as K5, K8, K14, E-cadherin, P-cadherin, and plakoglobin in metastatic breast CTC clusters [116]. Accordingly, knockdown of plakoglobin, a member of the catenin protein family and homologous to β-catenin, led to disaggregation of the CTC clusters, thereby compromising metastasis formation [104]. Also, disruption of K14 expression negatively affected the expression of key downstream effectors in metastatic niche remodeling and metastasis survival, leading to compromised efficiency in metastasis formation [105]. However, CTC clusters with predominant hybrid E/M or fully mesenchymal features have been observed in human colon, prostate, lung, and breast cancer patients [93,99,100,101]. At least one-third of cancer cells from within CTC clusters derived from colon cancer patients were negative for cytokeratin expression [100]. In breast cancer, CTC clusters show shifts in their EMT status according to treatment modalities with predominant mesenchymal expression patterns during cancer progression and/or in refractory disease [93] (Figure 3). This dynamic EMT profile allows for cellular plasticity and adaptation to the diverse cellular contexts encountered by CTCs during dissemination and metastasis formation, and to different treatments regimes. The latter is also of relevance for the use of prognostic epithelial markers of CTCs likely to fail to detect cancer cells that have undergone EMT. Additional mesenchymal CTCs markers are needed for more accurate prognostic studies [117].

As mentioned above, alternative EMT-programs accounts for different CTC phenotypes. Aiello et al. (2018) suggested that single CTCs arise from cancer cells that have completed a full EMT-program, whereas tumor cells characterized by partial EMT tend to present as clusters resulting from collective migration [34]. However, it has been demonstrated that, next to those mainly composed by CTCs, cell clusters have been isolated from colon cancer patients which consist of endothelial cells without any genetic aberrations found in their matched primary tumor of origin. These cell clusters were positive for both epithelial and mesenchymal markers and are thought to result from the direct release of clusters from the tumor vasculature due to impaired neo-angiogenesis [118]. 

## 6. Partial EMT, Collective Cell Migration, and Metastasis: Therapeutic strategies

Metastasis formation involves the successful completion of a sequential series of challenging steps. Phenotypic plasticity refers to the key feature of the metastasizing cancer cell to adapt to the environment where it resides through reversible changes of its cellular identity [13]. This ‘*Dr. Jekyll and Mr. Hide*’ feature of migrating cancer cells is controlled by epigenetic mechanisms which regulate E-to-M and M-to-E transitions (EMT and MET) [14]. However, EMT cannot be regarded as a binary process as it generates hybrid E/M cancer cells encompassing a range of intermediate stages. Partial EMT has been correlated with collective cell migration and with the presence of CTC clusters with enhanced metastatic potential in the peripheral blood of cancer patients [119]. Moreover, the unaffected CTC clusters multiplicity upon chemotherapy is indicative of treatment failure in colorectal cancer [120]. From this perspective, the elucidation of the underlying intrinsic and extrinsic mechanisms is bound to lead to the development of novel therapeutic and even preventive strategies based on the targeting of cell–cell and/or cell-matrix interactions and the disruption of CTC clusters. Gkountela et al. (2019) tested a library of approximately 2500 FDA-approved compounds and identified Na^+^/K^+^ ATPase inhibitors able to disaggregate derived CTC clusters derived from breast cancer patients into single cells. Mechanistically, Na^+^/K^+^ ATPase inhibition in tumor cells leads to an increase of intercellular Ca^2+^ concentration and to the consequent inhibition of formation of cell–cell junctions. In an in vivo xenograft model using NSG mice injected with patient-derived breast cancer cells in their fat pad, treatment with the Na^+^/K^+^ ATPase inhibitor ouabain resulted in a marked reduction of CTC cluster formation together with the increase of single CTC multiplicity. Although the size of the primary tumor was unaffected upon ouabain treatment, the overall number of metastatic lesions, corresponding to the number of CTC clusters, was reduced [107]. 

An alternative approach towards the development of therapeutic strategies based on CTC clusters may be represented by inhibition of platelet function. Platelets make integral part of CTC clusters where they are thought to protect the cancer cells from shear stress and immune attacks [121]. Acetylsalicylic acid (i.e., aspirin) inhibits platelet function by acetylation of cyclooxygenase (COX) thereby preventing arachidonic acid (and prostaglandin) production and consequently resulting in irreversible inhibition of platelet-dependent thromboxane formation. Based on this, aspirin has been employed as an anticoagulant for the prevention of thrombosis [122]. In experimental cancer models and clinical trials, inhibiting the interaction between cancer cells and platelets have been shown to hamper tumor cell survival, growth and metastasis formation [123,124]. 

Notwithstanding the above promising and innovative therapeutic strategies based on CTC clusters, their allegedly high degree of plasticity—as a mechanism to escape targeted treatment—is also likely to result in therapy resistance. Nonetheless, future research towards the identification of novel therapeutic targets to lower the risk of CTC cluster formation is expected to improve the efficacy of cancer treatment in the long run. 

## 7. Final Remarks and Conclusions

EMT contributes to a considerable degree of cellular heterogeneity in both primary tumors and metastatic lesions as it affects a broad spectrum of cellular functions beyond the transitions between epithelial and mesenchymal states associated with enhanced invasive and metastatic abilities. Changes in stem cell behavior, escape from apoptosis and senescence, ECM and tumor-microenvironment remodeling, and resistance to cytotoxic treatments are only a few among the broad spectrum of downstream EMT effectors which contribute to intra-tumor cell heterogeneity with profound implications for cancer therapeutics, especially in the decade of personalized treatments [11].

In fact, EMT is thought to play key roles in each and every step of the metastatic cascade including intra- and extravasation [125], and the colonization of distant organ sites [126,127]. For the sake of brevity, these latter aspects are not discussed in this review. The observed broad spectrum of EMT effectors may well reflect the pleiotropic functional roles of the EMT-TFs such as ZEB1 [128] that go well beyond the E to M (and vice versa) trans-differentiation, and include angiogenesis, remodeling of the tumor microenvironment, immune escape, mechanotransduction, and possibly many more. 

The identification and elucidation of the complex network of intrinsic and extrinsic mechanisms driving EMT at “just-right” (E/M) levels to trigger collective migration, generate CTC clusters and successfully metastasize distant organ sites represent the major future research challenge in the translation of our fundamental understanding of metastasis into therapy. From this perspective, single-cell epigenetic and transcriptomic analysis will provide powerful approaches to address this challenge. These high-resolution techniques will be key to elucidate the heterogeneous composition of malignancies including the identification of distinct and rare cell types arising transiently in time and at specific locations within tumors. Moreover, single-cell profiles will help to investigate the variability among individuals, disease states, microenvironments, and treatment history.

Overall, the realization of the importance of epigenetics and the elucidation of the mechanisms underlying transient changes in the cellular identity of individual circulating and metastasizing tumor cells will lay the basis for the development of novel treatment modalities. These will complement the current ‘personalized cancer medicine’ mainly directed at somatic gene mutations arisen at the primary site and unlikely to be rate-limiting in the clinical management of a more advanced malignant disease. 

## Figures and Tables

**Figure 1 cancers-11-01368-f001:**
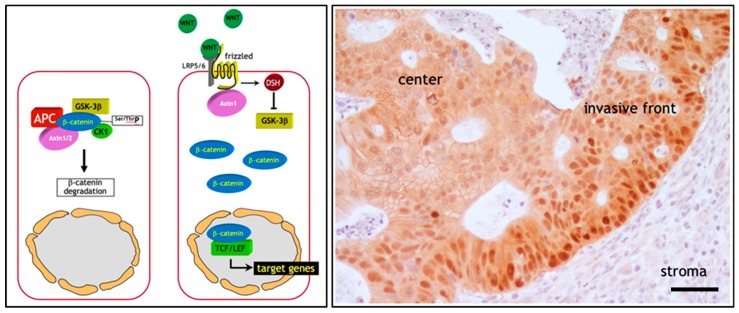
The (**a**) Wnt/β-catenin signal transduction pathway and the (**b**) β-catenin paradox in colon cancer. (**a**) Illustration of the canonical Wnt signaling in homeostasis. Left panel: In the absence of Wnt ligands, intracellular β-catenin levels are controlled by a destruction complex encompassing protein phosphatase 2A (PP2a), glycogen synthase kinase 3 (GSK3β) and casein kinase 1α (CK1α), adenomatous polyposis coli (APC), and Axin1/2. This complex binds and phosphorylates β-catenin at serine and threonine residues, thereby targeting it for ubiquitination and proteolytic degradation by the proteasome. Right panel: In presence of Wnt, co-activation of the Frizzled and LRP5/6 (low-density lipoprotein receptor-related proteins) receptors prevents the formation of the destruction complex leading to the stabilization and consequent translocation of β-catenin from the cytoplasm to the nucleus. Here, β-catenin interacts with members of the TCF/LEF family of transcription factors and modulates the expression of a broad spectrum of Wnt downstream target genes. Adapted from [24]. (**b**) The β-catenin paradox in colon cancer. β-catenin IHC analysis of the invasive front of a colon carcinoma show marked nuclear β-catenin accumulation in the proximity of the stromal microenvironment. In contrast, the majority of tumor cells localized inside the tumor mass are characterized by membrane-bound and cytoplasmic β-catenin staining. Scale bar: 50 µm.

**Figure 2 cancers-11-01368-f002:**
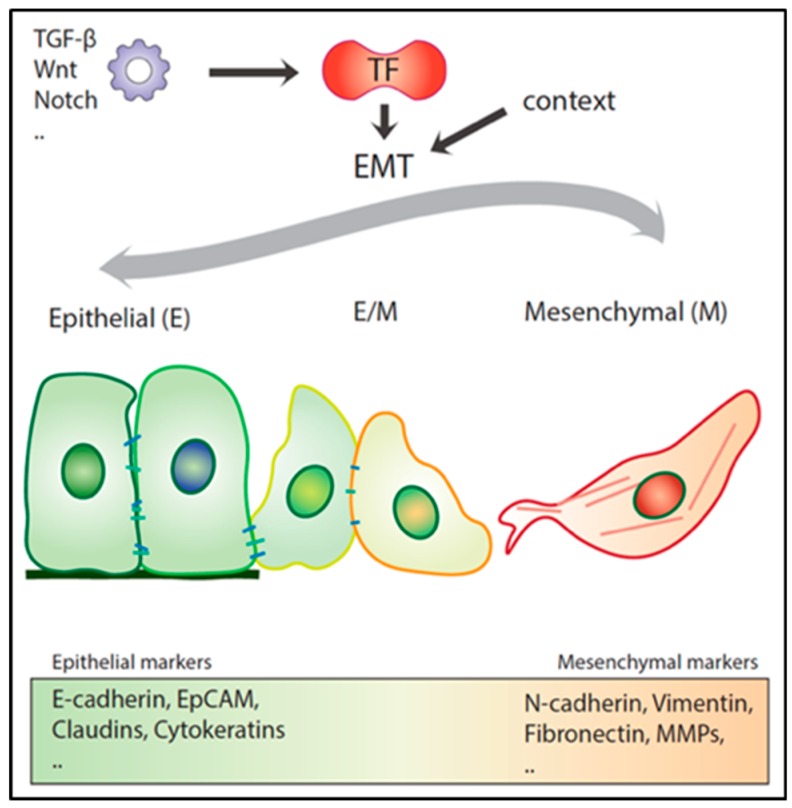
Epithelial to mesenchymal transition (EMT). Schematic overview of epithelial (E) cells transitioning to mesenchymal (M) phenotypes through an intermediate E/M state, and vice versa. EMT can be induced by various stimuli and is dependent on the environmental context.

**Figure 3 cancers-11-01368-f003:**
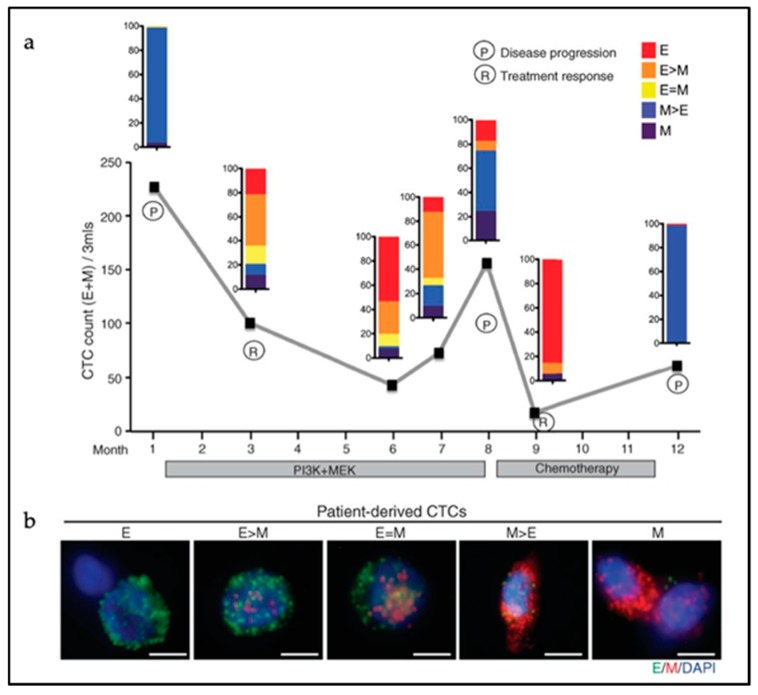
EMT features in single circulating tumor cells (CTCs) and CTC clusters from a metastatic breast cancer patient. (**a**) Longitudinal monitoring of EMT features in CTCs. The *Y*-axis indicates the number of CTCs per 3 mL of blood. The patient was monitored over time (*X*-axis) during treatment with inhibitors targeting the PI3K and MEK pathways (months 1–8), followed by chemotherapy with Adriamycin (8–12). The color-coded quantification bars indicate the EMT status of the CTCs based on RNA-ISH (in situ hybridization) analysis at each indicated time point. P = disease progression; R = tumor response. (**b**) RNA-ISH analysis of EMT markers in CTCs derived from patients with metastatic breast cancer. Green dots represent epithelial (E) and red marks mesenchymal (M) markers. Scale bar: 5 µm. Adapted from [93].

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
