# Peer review of "Cell Heterogeneity and Phenotypic Plasticity in Metastasis Formation: The Case of Colon Cancer"

_cancers, 2019, doi:10.3390/cancers11091368_

Round 1

Reviewer 1 Report

The review by Teeuwssen and Fodde addresses a very timely and interesting topic. The manuscript is well written and highlights some very important concepts. The only major concern I have relates to the fact that there is some kind of a discrepancy between the title and the content. According to the title the review is supposed to focus on colon cancer, yet the majority of the studies highlighted in the text are from other tumor entities and it is not clear whether the respective findings can be recapitulated in colon cancer as well. On the other hand several important studies relating to colorectal cancer cell plasticity, heterogeneity and metastasis formation are not addressed. Specifically, in vivo data on miRNA, transcription factors in EMT, the role of stem cells in invasion and metastasis, heterogeneity of different cell types (particularly stromal cells such as CAFs and T cells) or the relevance of the intestinal microbiome is not addressed. I would strongly recommend to focus more specifically on colorectal cancer or alternatively to change the title.

Author Response

Reviewer #1. The review by Teeuwssen and Fodde addresses a very timely and interesting topic. The manuscript is well written and highlights some very important concepts. The only major concern I have relates to the fact that there is some kind of a discrepancy between the title and the content. According to the title the review is supposed to focus on colon cancer, yet the majority of the studies highlighted in the text are from other tumor entities and it is not clear whether the respective findings can be recapitulated in colon cancer as well. On the other hand, several important studies relating to colorectal cancer cell plasticity, heterogeneity and metastasis formation are not addressed.

Authors’ reply: We thank reviewer #1 for his/her kind words. Indeed, it is difficult to exclusively base ourselves on the admittedly limited colon cancer-related literature when addressing the general issue of phenotypic plasticity and that is why we felt compelled to refer to other tumor types and in vivo/in vitro models. To the best of our knowledge, we have covered all the available literature on colon cancer but clearly cannot exclude having missed some. Nevertheless, we have slightly modified the title to highlight the focus on the issue of phenotypic plasticity using colon cancer, among others, as a ‘test case’.

Specifically, in vivo data on miRNA, transcription factors in EMT, the role of stem cells in invasion and metastasis, heterogeneity of different cell types (particularly stromal cells such as CAFs and T cells) or the relevance of the intestinal microbiome is not addressed. I would strongly recommend to focus more specifically on colorectal cancer or alternatively to change the title.

Authors’ reply: We do appreciate the relevance of the topics mentioned by the reviewer but again, to focus on these aspects in the absence of specific and high-impact publications, would mainly result in either speculation or in “future perspectives”. As for miRNA, EMT-TFs, and cellular heterogeneity, we believe that we did address these issues in the introduction on EMT and collective cell migration. Also in these cases, the available literature comes from tumor types other than colon cancer (mainly pancreas and breast). Nevertheless, in the revised version, we have tried to include as much as possible colon cancer-related literature especially when it comes to the functions of CAFs in invasion/metastasis, including a recent paper on the role of CAF-derived exosomes in the activation of Wnt signaling and EMT in colon cancer. Lastly, we included a brief discussion on the role of the intestinal microbiome in EMT. In the revised version, the above-mentioned additions have been highlighted in red.

Reviewer 2 Report

This review article summarizes and discusses the current literature about the role of cancer cell heterogeneity and plasticity in cancer progression and metastasis, thereby focusing on colorectal cancer. It is a timely, well composed and structured review senior-authored by an experienced, world leading expert in the field.

The authors summarize and discuss the topic in a well-written and well-composed manuscript, which is also understandable by non-experts in the field. The selection of references is well-balanced. The review gives an overview on the role of tumor heterogeneity and plasticity for tumor progression in general (introductory chapter 1). Then it focusses the overall concept on the crucial activation of Wnt-signaling and accumulation of beta-catenin in colon cancer by using the beta-catenin paradox as prominent example (chapter 2). This directly leads to the well-known (but also highly debated) EMT-MET switches as a major program underlying heterogeneity/plasticity (in a transient and partial fashion) (chapter 3). Of note the current debate on this topic is not excluded. Finally they also discuss that the modes of tumor cell migration (chapter 4) and the generation of CTCs (chapter 5) are controlled by partial EMT activation.

Thus, the topic is of high interest for cancer biology as well as of high translational and clinical relevance. A review covering this hot field is therefore of immediate interest for cell biologists, cancer researchers and clinical oncologists.

In my view the manuscript is acceptable for publication.

I only have one minor suggestion to the chapter 2 (page 3). The author may include a paper extending the beta-catenin paradox to metastases, showing that the nuclear beta-catenin accumulation is reversed in colon cancer metastases (Brabletz et al, PNAS, 98,10356, 2019). This paper first proposed EMT (de-differentiation)/MET (re-differentiation) and the underlying phenotypic plasticity as driver of metastasis, which therefore cannot be explained by accumulating genetic alterations alone.

Author Response

Reviewer #2. This review article summarizes and discusses the current literature about the role of cancer cell heterogeneity and plasticity in cancer progression and metastasis, thereby focusing on colorectal cancer. It is a timely, well composed and structured review senior-authored by an experienced, world leading expert in the field.

The authors summarize and discuss the topic in a well-written and well-composed manuscript, which is also understandable by non-experts in the field. The selection of references is well-balanced. The review gives an overview on the role of tumor heterogeneity and plasticity for tumor progression in general (introductory chapter 1). Then it focusses the overall concept on the crucial activation of Wnt-signaling and accumulation of beta-catenin in colon cancer by using the beta-catenin paradox as prominent example (chapter 2). This directly leads to the well-known (but also highly debated) EMT-MET switches as a major program underlying heterogeneity/plasticity (in a transient and partial fashion) (chapter 3). Of note the current debate on this topic is not excluded. Finally, they also discuss that the modes of tumor cell migration (chapter 4) and the generation of CTCs (chapter 5) are controlled by partial EMT activation.

Thus, the topic is of high interest for cancer biology as well as of high translational and clinical relevance. A review covering this hot field is therefore of immediate interest for cell biologists, cancer researchers and clinical oncologists. In my view the manuscript is acceptable for publication.

I only have one minor suggestion to the chapter 2 (page 3). The author may include a paper extending the beta-catenin paradox to metastases, showing that the nuclear beta-catenin accumulation is reversed in colon cancer metastases (Brabletz et al, PNAS, 98,10356, 2019). This paper first proposed EMT (de-differentiation)/MET (re-differentiation) and the underlying phenotypic plasticity as driver of metastasis, which therefore cannot be explained by accumulating genetic alterations alone.

Authors’ reply: We thank reviewer #2 for his/her very positive words on our manuscript. We are also grateful for the suggestion on the missed paper by Brabletz et al. that we have added and commented in the revised manuscript (marked in red).

Reviewer 3 Report

Overall a well written review with extensive focus on EMT and its relation to cancer heteroeneity and phenotypic plasticity.

Even though epigenetic regulation is not the main focus of the review, a more incisive discussion of the epigenetic regulation of EMT could perhaps be made between lines 55 and 61 prior to the main content of the review.

Of recent interest are the effect of mechanosignling on EMT. Given that mechanosignaling can be heterogeneous with the tumor microenvironment perhaps it can be discussion in section 3, between lines 151 and 182, especially in the context of the extracellular matrix interaction.

Additionally, given the advent of single-cell transcriptomic and epigenetic techniques, the authors can further comment on how such techniques can contribute to understanding the problems they highlighted.

Author Response

Reviewer #3. Overall a well written review with extensive focus on EMT and its relation to cancer heterogeneity and phenotypic plasticity. Even though epigenetic regulation is not the main focus of the review, a more incisive discussion of the epigenetic regulation of EMT could perhaps be made between lines 55 and 61 prior to the main content of the review.

Of recent interest are the effect of mechanosignaling on EMT. Given that mechanosignaling can be heterogeneous with the tumor microenvironment perhaps it can be discussion in section 3, between lines 151 and 182, especially in the context of the extracellular matrix interaction.

Additionally, given the advent of single-cell transcriptomic and epigenetic techniques, the authors can further comment on how such techniques can contribute to understanding the problems they highlighted.

Authors’ reply: We are grateful to reviewer #3 for the overall positive evaluation.

As for epigenetics, we have given a more global describing chromatin remodelers and histone modifications in the introductory section and, as such, this part was deliberately not as incisive as the reviewer would have liked. More detailed descriptions of the epigenetic regulation of EMT have now been added in the revised manuscript although, as the reviewer also pointed out, this is not the main focus of the review. Moreover, as also discussed in our rebuttal to reviewer #1, the colon cancer-specific literature on this subject is rather limited. 

As for mechano-sensing, we have now enumerated the relevant EMT-inducible factors and have included new references which highlight the role of cell shearing and rigidity of the ECM in EMT induction (marked in red in the revised version). We would like to thank the reviewer for this useful suggestion.

In the very last paragraph, we have added a short conclusive discussion on the future role of single cell omics technologies to further elucidate the complexity of EMT.

Round 2

Reviewer 1 Report

My concerns have been adequately addressed.